# Rapid Antigen Tests during the COVID-19 Era in Korea and Their Implementation as a Detection Tool for Other Infectious Diseases

**DOI:** 10.3390/bioengineering10030322

**Published:** 2023-03-03

**Authors:** Kristin Widyasari, Sunjoo Kim

**Affiliations:** 1Gyeongsang Institute of Health Sciences, Gyeongsang National University, Jinju 52727, Republic of Korea; 2Department of Laboratory Medicine, College of Medicine, Gyeongsang National University, Jinju 52727, Republic of Korea; 3Department of Laboratory Medicine, Gyeongsang National University Changwon Hospital, Changwon 51472, Republic of Korea

**Keywords:** rapid antigen test, antigen, infectious agent, clinical performance, lateral immunoassay

## Abstract

Rapid antigen tests (RATs) are diagnostic tools developed to specifically detect a certain protein of infectious agents (viruses, bacteria, or parasites). RATs are easily accessible due to their rapidity and simplicity. During the COVID-19 pandemic, RATs have been widely used in detecting the presence of the specific SARS-CoV-2 antigen in respiratory samples from suspected individuals. Here, the authors review the application of RATs as detection tools for COVID-19, particularly in Korea, as well as for several other infectious diseases. To address these issues, we present general knowledge on the design of RATs that adopt the lateral flow immunoassay for the detection of the analyte (antigen). The authors then discuss the clinical utilization of the authorized RATs amidst the battle against the COVID-19 pandemic in Korea and their role in comparison with other detection methods. We also discuss the implementation of RATs for other, non-COVID-19 infectious diseases, the challenges that may arise during the application, the limitations of RATs as clinical detection tools, as well as the possible problem solving for those challenges to maximize the performance of RATs and avoiding any misinterpretation of the test result.

## 1. Introduction

Infectious agents such as viruses, bacteria, fungi, and parasites can cause infections in humans. A proper diagnosis by detecting the correct pathogens that caused the infection improves the effectiveness of treatments and is essential for selecting the correct control measures and may be helpful to avoid complications for the infected patient. Moreover, a rapid diagnosis may prevent disease transmission from infected individuals to others [1]. The methods of detection vary, including the cultivation of suspected pathogens (bacteria and fungi) from samples on a growth medium, isolation of viruses in cell cultures, and identification of the pathogenic agents through the detection of their antigens (specific proteins) or genetic materials [2]. In addition, the host immune response that develops during the course of illness can also be used for the diagnosis of infectious diseases [2].

Amid the battle against the COVID-19 pandemic, the rapid identification of either symptomatic or asymptomatic cases is crucial to prevent the further spread of COVID-19. Moreover, efficient testing in the early stages of infections is necessary to distinguish COVID-19 from other diseases. A thorough inspection and identification will minimize unnecessary quarantines of COVID-19-negative individuals while, at the same time, preventing further transmission by positive individuals. Additionally, proper identification during the early stages of infection permits healthcare workers to provide immediate care and treatments for high-risk individuals in developing more serious complications of COVID-19.

The rapid development in science and technology brings new insight into an innovative diagnosis that allows for point-of-care testing and real-time diagnostics in patients with suspected diseases. Rapid antigen diagnostic tests (RADTs/RATs) enable the direct detection of a specific protein (antigen) of infectious agents in a specimen, e.g., nasal/nasopharyngeal swabs, oropharyngeal swabs, urine, blood, or saliva [3].

RATs specifically detect the protein of the target infectious agent; hence, they are specific to a certain infectious agent. However, RATs do not amplify the sample, so the test will only return a positive result when the amount of the infectious agent reaches a high level in a person’s body. This is in contrast to the polymerase chain reaction (PCR)-based test, which enables amplification of the pathogen’s genetic material and can therefore detect what are initially extremely small quantities of viruses [4]. Although the PCR test has a high degree of accuracy, the procedure is complex, the data analysis can be time-consuming, and an expert is required to conduct the sophisticated test. Meanwhile, RATs are considered relatively easy to handle and simple to use, do not require expertise, are suitable for point-of-care testing, are low-cost, and have a shorter turnaround time than other diagnostic platforms [5]. Due to these reasons, RATs are recommended to be used for an individual with symptoms but who cannot get access to timely PCR testing, for individuals who are in close contact with infected individuals, for the first screening of infected individuals before confirmation using a molecular-based test in a high-risk community, or as confirmatory testing after recovery from the infection [6]. Hence, RATs are considered a potential first-phase diagnostic tool that can be used as an alternative to the more complex and technically demanding assays.

In this review, we presented an overview of the development and clinical performance of RATs during the COVID-19 pandemic era, particularly the currently available RATs and those that have received authorization in Korea. We also discuss the implementation of RATs as a detection tool for other infectious diseases, the obstacles that may arise during the clinical application of RATs, the general limitation of the RATs, and the possible problem-solving that may be applied to maximize the performance of RATs based on the published literature.

## 2. Database Search Strategy

The inclusion criteria used for the literature search in this review included studies that discussed lateral flow assays for antigen detection, antigen tests for COVID-19, RATs that were commercially available and had received authorization in Korea, RATs for non-COVID-19 diseases, and the clinical performance of RATs. For this purpose, the authors searched and identified the relevant publications in the PubMed and Google Scholar databases, as well as online articles that covered the above-mentioned topics as of 2 February 2023 with the keywords “COVID-19 diagnosis”, “rapid antigen test”, “authorized RATs in Korea”, “detection of infectious diseases”, “antigen test for infectious diseases”, “sensitivity of RATs”, “specificity of RATs”, “clinical performance of RAT”, “antigen test for respiratory diseases”, “RAT for sexually transmitted infections”, and “limitation of RATs”.

## 3. Lateral Flow Immunoassay-Based Rapid Antigen Detection Test for COVID-19

RATs are commonly designed based on the lateral flow immunochromatographic assay (LFIA). The LFIA is a paper-based platform that uses nitrocellulose membranes, colored nanoparticles, and typically antibodies to detect the presence of one or more target analytes (usually antigens) in the sample [7]. The LFIA has a relatively low development cost, and it is easy to produce in a large volume; moreover, LFIA-based tests have a short turnaround time for assays, usually around 15–30 min. Hence, applications of LFIAs have spread to a variety of areas where quick tests are needed, including screening centers, hospitals, offices, and clinical laboratories. LFIA-based tests are also applied to a variety of biological samples including urine [8], saliva [9], serum [10], plasma, and whole blood [11].

The LFIA-based RAT was first developed in the 1980s for the identification of group A streptococcus (GAS) [12]. This test closely mimics the urine pregnancy test that gained wide acceptance due to its general simplicity, low cost, and comparatively high accuracy [13]. Later, RATs were also employed for the detection of several other pathogens that became worldwide concerns, such as influenza and malaria [12]. Following the outbreak of SARS-CoV-2 by the end of 2019, RATs have played a crucial role in aiding the initial diagnosis of SARS-CoV-2. Given that RATs use the principle of the LFIA, the assessment will return qualitative results that can be seen with the naked eye, making the interpretation of the test results relatively easier for the user. In most cases, the manufacturers provide not only all materials required for performing the test, but also simple instructions with pictures that depict the test results (positive, negative, and invalid). Additionally, LFIA-based RATs have a shelf-life of up to two years and do not require special storage conditions (generally at room temperature) [13].

The RAT for COVID-19 can detect the specific SARS-CoV-2 antigen (nucleocapsid (N) protein) in respiratory samples. The test is conducted by collecting a sample, mostly from the nasal cavity of each nostril, or the saliva, by using a swab (nasopharyngeal, oropharyngeal); subsequently, the sample is mixed with a specific buffer before being applied to the test cassette [14]. When the sample–buffer mixtures are added onto the sample pad of the test cassette, the N protein will bind with a colloidal gold-labeled antibody that is specific to the SARS-CoV-2 N protein on the conjugation pad and will later form an antibody–antigen complex. Apart from the gold-labeled SARS-CoV-2 N protein’s specific antibody, the conjugation pad also contains colloidal gold-labeled chicken IgY antibodies that will be captured by goat anti-chicken IgY on the control line, which serves as a procedural control [15,16].

Subsequently, the antibody–antigen complex and the chicken IgY antibodies (control) will migrate via diffusion across the nitrocellulose membrane and come into proximity with the test (T) and control (C) lines. The SARS-CoV-2 N protein–antibody complex will bind to the anti-nucleocapsid protein from different epitopes on the T line and form an antibody–antigen–antibody complex, resulting in the colored line appearing on the T line. The excess sample that does not bind to the antibody on the T line will diffuse and bind to the goat-anti-chicken IgY, leading to the development of the colored line on the C line [17] (Figure 1).

## 4. Antigen Tests Amid the Battle against SARS-CoV-2 in Korea

### 4.1. Authorized Devices for SARS-CoV-2 Antigen Diagnostic Testing

Over the three years of the COVID-19 pandemic, rapid antigen tests have become part of daily life. RATs are considered a complement to PCR tests, hence being widely employed in the first phase of screening. Additionally, RATs are cheap and easy to use, enabling point-of-care testing with a short execution time (15–30 min) [5], making RATs a favorable SARS-CoV-2 detection tool. To date, numerous RATs are commercially available and have been used worldwide. In Korea, around 12 RATs have received approval from the Korean Ministry of Food and Drug Safety and are currently available in clinical practice (Table 1). Meanwhile, the U.S. Food and Drug Administration (FDA) has issued emergency use authorization for a total of 59 SARS-CoV-2 RATs [18].

The commercially available RATs are supplemented with information on their sensitivity and specificity in detecting SARS-CoV-2. However, the majority of data provided by the manufacturers come from clinical trials on people with high viral loads (true infections), thus allowing the RATs to appear highly sensitive and specific [4]. Real-world conditions, however, demonstrate a high variability of viral loads even among infected individuals, depending on the infection period and disease severity. Hence, a proper evaluation of the sensitivity and specificity of commercially available RATs under real-world conditions is necessary to avoid any misinterpretation of the test results.

Clinical performance evaluation demonstrates divergence of the sensitivity and specificity of RATs depending on the product, the sample that is being used, and the period of the test [5,33,34,35]. Although not all commercially available COVID-19 RATs have been sufficiently evaluated under real-world conditions, in the following section, we discuss seven of the twelve authorized RATs in Korea that have been clinically evaluated by several independent study groups in and outside of Korea. As for the other five authorized RATs in Korea, their clinical evaluation report is very limited. The sensitivity and specificity information of these five RATs are limited only to those provided by the manufacturers.

#### 4.1.1. STANDARD^TM^ Q COVID-19 Ag Home Test

The STANDARD^TM^ Q COVID-19 Ag Home Test is one of the immunochromatography rapid antigen detection kits produced by SD BIOSENSOR, Gyeonggi-do, Korea. Clinical performance evaluation conducted by several independent groups demonstrated the varying sensitivity of this RAT according to the infection periods. The sensitivity was at its highest when the assessment was conducted in the early days (<5 days) of the disease. Within 5 days of symptom onset, the sensitivity of the STANDARD^TM^ Q COVID-19 Ag Home Test ranged from 66.7% to 100% [36,37]. At a median of 3 days from the onset of symptoms, the STANDARD^TM^ Q COVID-19 Ag Home Test demonstrated a sensitivity and specificity of 98.33% and 98.7%, respectively [38].

Additionally, this RAT also demonstrated strong agreement with reverse transcription PCR (RT-PCR) during the first 5 days of illness [37] and excellent agreement with viral cultures, suggesting that the STANDARD^TM^ Q COVID-19 Ag Home Test may be used as a marker of contagiousness [39].

The viral load also affects the performance of the STANDARD^TM^ Q COVID-19 Ag Home Test. In patients with a high viral load (>7.0 log_10_ RNA SARS-CoV-2/swab), the STANDARD^TM^ Q COVID-19 Ag Home Test demonstrated a high sensitivity of 95.7% when the test was self-conducted, with a sensitivity as high as 100% when the sample collection and tests were conducted by professional healthcare workers [40].

Subsequently, the cycle of threshold (Ct) values, which are often related to the viral load, also contributed to the variation in the sensitivity and specificity of the STANDARD^TM^ Q COVID-19 Ag Home Test. The sample with Ct values of ≤25 demonstrated a sensitivity of up to 100%, whereas this sensitivity decreased when the assessment was conducted on samples with Ct values > 25 (sensitivity of 73.33%) [34]. Another study on the significance of Ct values to the detection rate of the STANDARD^TM^ Q COVID-19 Ag Home Test also demonstrated comparable results, where the detection rate reached up to 100% when used on the samples with Ct values ≤ 25, decreasing to 38.5% for Ct values of 28–30, 26.7% for Ct values of 30–35, and 9.1% for Ct values > 35 [41].

#### 4.1.2. GenBody COVID-19 Ag

The GenBody COVID-19 Ag is an immunochromatographic rapid antigen test intended for qualitative determination of SARS-CoV-2 infection from nasal and nasopharyngeal swab specimens of the suspected individuals [42]. The clinical validation of the GenBody COVID-19 Ag in the European adult population demonstrated 100% specificity for nasopharyngeal and nasal swab specimens, and a sensitivity of 94.17% and 97.09%, respectively [42]. Tests on the samples with higher viral loads yielded significantly better results. In concordance with this report, a retrospective evaluation of the 130 residual nasopharyngeal swab specimens transferred in a viral transport medium revealed a sensitivity and specificity of 90.0% and 98.0%, respectively [43]. Subsequently, the RATs used for assessment of the 200 symptomatic and asymptomatic nasopharyngeal swabs, collected onsite, demonstrated a significantly higher sensitivity and specificity of 94.00% and 100%, respectively. This study also revealed that the sensitivity and specificity of the GenBody COVID-19 Ag were significantly higher for samples with lower Ct values.

#### 4.1.3. STANDARD^TM^ F COVID-19 Ag FIA

The STANDARD^TM^ F COVID-19 Ag FIA (SD Biosensor, Gyeonggi-do, Korea) is a fluorescent immunoassay for the qualitative detection of SARS-CoV-2 specific N protein from the nasopharyngeal specimens of the suspected individuals. Clinical performance of the STANDARD^TM^ F COVID-19 Ag FIA in 663 specimens from non-repetitive patients demonstrated a clinical sensitivity and specificity of 84.0% and 99.6%, respectively, with the kappa index agreement between RT-PCR and the RAT being 0.89 [44]. In another study, however, the STANDARD^TM^ F COVID-19 Ag FIA demonstrated a low sensitivity, although the specificity, positive predictive value, and negative predictive value were considered high (38%, 99%, 96.2%, and 72%, respectively). The overall diagnostic accuracy of this RAT according to this study is 75.7% [45].

#### 4.1.4. BIOCREDIT COVID-19 Ag

BIOCREDIT COVID-19 Ag is a rapid chromatographic immunoassay produced by Rapigen Inc., Gyeonggi-do, Korea, which is intended for the qualitative detection of SARS-CoV-2 N protein from nasal swab samples. Clinical performance evaluation using samples from individuals with COVID-19 infection and/or suspected infection in Hong Kong demonstrated a low concordance of BIOCREDIT COVID-19 Ag with RT-PCR. BIOCREDIT COVID-19 Ag was reported to only be able to detect between 11.1% and 45.7% of RT-PCR-positive samples. Moreover, this RAT was also reported to be 10^3^-fold less sensitive than viral cultures [46].

The RAT sensitivity of the mentioned study, however, demonstrated a significant increase of up to 81.8% when BIOCREDIT COVID-19 Ag was used to evaluate the sample with a higher viral load. A similar result was also reported following the clinical assessment of BIOCREDIT COVID-19 Ag in Egypt. BIOCREDIT COVID-19 Ag demonstrated an overall sensitivity of 43.1% in the analysis of samples with a lower viral load (Ct value range: 15.8–32.3) but a significantly higher sensitivity (93.4%) in the assessment of samples with a higher viral load (Ct value range: 15.8–25.5). All false negatives in the RAT corresponded to specimens with Ct values > 28 [47].

Comparison analysis of the performance of several RATs, including BIOCREDIT COVID-19 Ag, in Uganda demonstrated that BIOCREDIT COVID-19 Ag had a specificity of ≥97% for the assessment of samples with a lower Ct value (≤29), and none of the evaluated RATs had a sensitivity of more than 80% at Ct values higher than 29 [48].

#### 4.1.5. Humasis COVID-19 Ag Test

The Humasis COVID-19 Ag Test (Humasis, Gyeonggi-do, Korea) is an in vitro diagnosis kit intended for the detection of the specific SARS-CoV-2 antigen from nasal or nasopharyngeal swab specimens. The retrospective study that was carried out at the Croatian Institute of Public Health demonstrated satisfactory performance of the Humasis COVID-19 Ag Test, particularly in younger patients. From a total of 2490 symptomatic patients, the RAT gave 953 positive results and 1537 negative results. Among the RAT negative results, 266 samples were confirmed to be false negatives by RT-PCR. Thus, the overall efficacy of the Humasis COVID-19 Ag Test according to this study was 82.69% [49]. In addition, on-field performance evaluation of 10 commercially available SARS-CoV-2 RATs, including the Humasis COVID-19 Ag Test, found the sensitivity and specificity of this RAT to be 80% and 79%, respectively [50].

#### 4.1.6. CareStart^TM^ COVID-19 Antigen Home Test

The CareStart^TM^ COVID-19 Antigen Home Test is a lateral flow immunochromatographic assay produced by Access Bio, Somerset, NJ, U.S.A., for the qualitative assessment of the nucleocapsid protein of SARS-CoV-2 in nasopharyngeal or anterior nasal swab specimens from individuals with suspected SARS-CoV-2 infection. Performance evaluation of the CareStart^TM^ COVID-19 Antigen Home Test demonstrated a sensitivity/specificity of 84.8%/97.2% and 85.7%/89.5% in adults and children, respectively, within 5 days of symptom onset. Ct values ≤ 25 returned a sensitivity of 96.3%, whilst sensitivities of 79.6% and 61.4% were obtained in the assessment of samples with Ct values ≤ 30 and ≤35, respectively [51].

The performance of the CareStart^TM^ COVID-19 Antigen Home Test, however, decreased when the assessment was conducted on samples from asymptomatic individuals. This condition is likely due to the low viral load in the asymptomatic individuals, hence making the sensitivity and specificity of the RAT for this batch of samples less satisfying than the one for the assessment in symptomatic individuals.

A comparably high sensitivity and specificity of the CareStart^TM^ COVID-19 Antigen Home Test were also reported in another study. The sensitivity of the RAT for the assessment of nasopharyngeal samples from symptomatic individuals was reported to be as high as 84.57%. Meanwhile, the specificity was as high as ≥95%. Additionally, the CareStart^TM^ COVID-19 Antigen Home Test was reported to be able to detect patients with different variants of SARS-CoV-2 [52].

Another study that focused on the assessment of asymptomatic participants showed that the CareStart^TM^ COVID-19 Antigen Home Test had a high specificity (99.5%) and moderate sensitivity (49.0%). The Ct values for samples that tested positive using the CareStart^TM^ COVID-19 Antigen Home Test were significantly lower than those of samples that tested negative [53], implying that the Ct values of samples affect the performance of the CareStart^TM^ COVID-19 Antigen Home test.

#### 4.1.7. Panbio^TM^ COVID-19 Ag Self-Test

The Panbio^TM^ COVID-19 Ag Self-Test is an antigen test kit produced by Abbott Laboratories, North Chicago, IL, U.S.A., that is intended for the qualitative assessment of the SARS-CoV-2 antigen in human nasal swab samples from individuals with infection or suspected SARS-CoV-2 infection. Performance evaluation of the Panbio^TM^ COVID-19 Ag Self-Test for the assessment of samples from a total of 3014 asymptomatic individuals in Canada demonstrated moderate concordance (54.5%) with RT-PCR. All RAT-positive tested samples were also confirmed as positive by RT-PCR. The highest corresponding Ct value for the Panbio^TM^ COVID-19 Ag-positive samples was 28.48 [54]. Subsequently, the sensitivity and specificity of the Panbio^TM^ COVID-19 Ag Self-Test were reported to be higher when used for the assessment of samples from symptomatic individuals [55]. The sensitivity of the RAT when used for the assessment of symptomatic individuals ranged from 79.4% to 90.5%, whilst the specificity ranged from 98.9% to 100% [55,56,57]. These values are acceptable when compared to the thresholds suggested by the World Health Organization.

The days after symptom onset and the Ct values that represent viral loads play a crucial role in the sensitivity and specificity of RATs. Given that RATs assessment depends on the presence of the specific viral protein (antigen) for the detection, the viral loads in the samples are a determining factor for positive outcomes. Thus, a lower Ct value (<25) corresponds to a positive RAT result.

Subsequently, the symptoms are likely to develop due to the manifestation of a high accumulation of the virus. Hence, a RAT assessment conducted in symptomatic patients during the early days of symptom onset (<5 days) is likely to return a positive result with high sensitivity and specificity. On the contrary, an assessment conducted in individuals without symptoms, or before or long after symptom onset, will likely return a negative result (Figure 2).

### 4.2. RATs Are Complements of PCR for the Detection of SARS-CoV-2

In the three years since the first case of COVID-19 was reported, the disease has caused tremendous changes in our society. Multiple control measures, such as the administration of COVID-19 vaccines worldwide, mandatory mask policy, quarantine, and isolation policies, as well as the rolling out of COVID-19 tests for high-risk regions or populations, have been implemented to contain the pandemic. Among all efforts that have been made, COVID-19 testing plays a key role in containing and mitigating the COVID-19 pandemic by identifying the infected individuals, thus helping to prevent further person-to-person transmission of SARS-CoV-2. Two main types of SARS-CoV-2 viral tests, i.e., nucleic acid amplification tests (NAATs) and antigen tests, are available worldwide. NAATs such as RT-PCR for SARS-CoV-2 are designed to detect and amplify the viral RNA, thus enabling the detection of even minuscule quantities of the virus. Meanwhile, antigen tests, often referred to as rapid antigen tests or RATs, detect the specific SARS-CoV-2 proteins (antigens). Unlike RT-PCR, the performance of RATs depends on the viral loads in the samples to determine the test results [58]. Adequate viral loads yield a positive test result, whilst insufficient viral loads will lead to negative results. Due to this condition, RATs may be less sensitive in the detection of an individual who is in the early stage of the incubation period. Nevertheless, compared to the RT-PCR test, RATs are relatively easy to use, enable point-of-care testing with shorter turnaround time, do not require expertise, and are cheaper, hence making them an affordable option even for regions with low resources (Table 2).

Although the RT-PCR test remains the gold standard for the detection of SARS-CoV-2, RATs have proved to aid the scaling up of testing capacities worldwide [59]. RATs are not intended to be a substitute for RT-PCR tests but a complement; thus, RATs should not be used to assess the prevalence of COVID-19 in the population but for the prevention of the further transmission of SARS-CoV-2.

Additionally, given that taking a RAT during the early incubation period before the onset of symptoms will likely yield negative results, it is suggested that RATs are performed as serial testing at a specified frequency during the incubation period until the onset of the early symptoms (e.g., every two days after contact with a person with COVID-19 infection).

## 5. RATs as a Detection Tool to Aid the Clinical Diagnosis of Other Infectious Diseases

During the COVID-19 pandemic era, RATs have emerged as an important alternative SARS-CoV-2 detection tool. Although RATs are reported to have lower sensitivities when used for the assessment of asymptomatic individuals or during the early incubation period, the application of RATs for symptomatic individuals and those who have a high (sufficient) viral load yields a high sensitivity [60,61]. Hence, proper knowledge of how and when to apply RATs for the alternative detection of suspected pathogens will bring a lot of benefits, especially for screening purposes, thus enabling the prevention of person-to-person transmission and offering an extra layer of protection in a comprehensive public health response.

The application of RATs as a detection tool to aid the clinical diagnosis is not only limited to COVID-19 disease. The application of RATs for the detection of pathogens including influenza viruses, respiratory syncytial virus (RSV), *Streptococcus pneumoniae*, *Streptococcus pyogenes*, dengue virus, malaria parasite, and infectious agents that cause multiple sexually transmitted infections has been studied, and reported by multiple groups. The ultimate goal of pathogen detection tools is to aid physicians in making a timely diagnosis and managing the patient properly. Thus, RATs with good sensitivity and specificity are good potential candidates for these purposes when applied at the proper time and in the appropriate procedures. In the sections below, we further discuss the clinical application of RATs for the detection of pathogens that cause diseases other than COVID-19.

### 5.1. Clinical Application of RATs for Respiratory Diseases Other than COVID-19

Acute respiratory infections other than COVID-19, caused by viruses, bacteria, or fungi, are the second leading causes of morbidity and mortality in children and adults [62]. Prompt and accurate detection of infectious agents is important so that the right intervention can be put in place to confine the infection and prevent further transmission of the disease. Clinical applications of antigen tests for the detection of the infectious agents that cause respiratory diseases have been studied before the outbreak of the COVID-19 pandemic. However, the real-world application of these RATs in healthcare centers or other institutions is not as prominent as that of RATs used for the detection of SARS-CoV-2.

The RATs for group A Streptococcus (*S. pyogenes*) (GAS) that cause pharyngitis appeared following the use of antigens as the analyte for assessment at the beginning of the 1980s [62]. Since then, several antigen tests that employ different methodologies, including the LFIA that is widely used in current RATs, have been generated. A review of the clinical performance of RATs for GAS provided summarized values for the sensitivity and specificity of RATs, with values of up to 85.6% (95% CI, 83.3–87.6%) and 95.4% (95% CI, 94.5–96.2%), respectively. According to this review, substantial heterogenicity in the sensitivity of RATs across the studies was presented; however, the specificity of the RATs remained more stable [63]. The turnaround time for detection was comparable to that for SARS-CoV-2, which is around 15–30 min [62].

Furthermore, RATs were also reported to be applied for the detection of two bacteria that caused a type of pneumonia, *Streptococcus pneumoniae* and *Legionella pneumophila*. *S. pneumoniae* is the most commonly identified infectious agent in community-acquired pneumonia (CAP), whereas *L. pneumophila*, a causative agent of uncommon but potentially serious pneumonia (legionnaires’ disease), is considered less frequently identified in CAP [64]. Clinical evaluation of four *S. pneumoniae* urinary antigen tests—the BinaxNOW *S. pneumoniae* Antigen Card (Abbott, Chicago, IL, USA), ImmuView *S. pneumoniae* and Legionella (SSI Diagnostica, Hillerød, Denmark), STANDARD F *S. pneumoniae* Ag FIA (SD Biosensor, Gyeonggi, South Korea), and Sofia *S. pneumoniae* FIA (Quidel Corporation, San Diego, CA, USA)—demonstrated a sensitivity ranging from 76.9% to 86.5%, and a specificity that ranged from 84.2% to 89.7%. No significant difference was found among the four test kits [65]. Correspondingly, a comparison of a novel antigen test kit, IMMONOCATCH^TM^
*Streptococcus penumoniae* (EIKEN CHEMICAL CO., Ltd., Tokyo, Japan), with two other test kits, Uni-Gold^TM^
*S. pneumoniae* (Trinity Biotech, Bray, Ireland) and BinaxNOW *S. pneumoniae* (Abbott Laboratories, North Chicago, IL, USA), demonstrated comparable results. The sensitivity and specificity of the IMMUNOCATCH were 73.2% and 98.9%, respectively, compared to the Uni-Gold; conversely, when compared to BinaxNOW, they were 97.6% and 98.8%, respectively [66]. Together, these studies suggest that the RAT is a useful tool for the detection of *S. pneumoniae*.

Subsequently, the RAT was also widely used for the detection of *L. pneumophila*. The clinical evaluation of the Legionella *K*-SeT (CORIS BioConcept, Gembloux, Belgium), an immunochromatographic assay for the detection of *L. pneumophila* from a urine sample, demonstrated sensitivity, specificity, positive predictive value, negative predictive value, and accuracy of 75.5%, 100%, 100%, 98.7%, and 98.9%, respectively [67]. Furthermore, a head-to-head comparison between the ImmuView *L. pneumophila* and *L. longbeachae* urinary antigen test (SSI Diagnostica A/S, Hillerød, Denmark) and the BinaxNOW *Legionella* urinary antigen card (Abbott Laboratories, North Chicago, IL, USA) also demonstrated high sensitivity and specificity of 96.0% and 100%, respectively, for both RATs when being used to detect the *L. pneumophila* from the urine samples [68].

Additionally, antigen detection tests were also widely used for the detection of *Mycobacterium tuberculosis*, bacteria that cause tuberculosis diseases. Tuberculosis is a highly transmitted disease that is also considered one of the major problems in the world. Evaluation of the commercially available RAT for tuberculosis in India demonstrated that the performance of the RAT depended on the bacterial load in the sample. A sample with a high bacterial load yielded a positivity of up to 100%, but with a lower bacterial load, the positivity decreased by up to 12.06% [69]. Another study on the antigen test for tuberculosis, however, reported the potential of two *Mycobacterium tuberculosis*-specific antigens, named ESAT-6 and CFP10, as analytes of RAT for tuberculosis. The combination of the ESAT-6 and CFP10 was found to be highly sensitive and specific for diagnosis, with a sensitivity of up to 73% and a specificity of up to 93% [70]. Correspondingly, the clinical performance evaluation of the tuberculosis RAT developed by the Pasteur Institute of Iran, named PrTBK, demonstrated 100% sensitivity, 89% specificity, and 92% accuracy in comparison with the culture method [71]. Furthermore, a study in 2019 reported the detection of tuberculosis lipoarabinomannan (LAM) antigen in urine samples of children with presumed TB. According to this study, the urinary LAM antigen testing enables the point-of-care diagnosis with relatively high sensitivity up to 73.2% in confirmed intrathoracic tuberculosis (ITTB) cases and 76% in confirmed lymph node tuberculosis (LNTB) cases. The specificity of the LAM-RAT assay in children with ITTB and those with LNTB was 92% and 93%, respectively [72].

Recently, the rapid development of detection technology has enabled the detection of more than a single infectious agent using a single RAT. QuickNavi-Flu+COVID19 Ag, a RAT that is intended for the detection of influenza viruses and SARS-CoV-2 simultaneously using nasopharyngeal or anterior nasal samples, was used to assess a total of 1510 nasopharyngeal samples and 862 anterior nasal samples in Japan. This study reported that the sensitivity and specificity of the antigen test were 80.9% and 99.8%, respectively, for the nasopharyngeal samples. Meanwhile, the sensitivity and specificity of the RATs for the assessment of anterior nasal samples reached 67.8% and 100%, respectively [73]. A comparative study on RATs that were able to detect multiple viruses simultaneously using a single kit was also reported in Korea. The STANDARD Q COVID/FLU Ag Combo Test was reported to demonstrate a sensitivity of 92.73% and a specificity of 99.49% for the detection of SARS-CoV-2, a sensitivity of 92.22% and a specificity of 100% for influenza A, and a sensitivity of 91.18% and a specificity of 99.49% for influenza B [14]. Collectively, given the simplicity and swiftness of their implementation, as well as the relatively good performance of these RATs for the detection of the numerous infectious agents that cause respiratory diseases, the future of RATs as detection tools to aid clinical diagnosis is highly promising.

### 5.2. Clinical Application of RATs for Dengue Fever

The NS1 antigen test is reported to be one of the RATs for the detection of specific dengue virus antigens in serum samples. An assessment of a total of 208 sera from patients suspected of having dengue virus infection in Malaysia reported that the performance of this RAT was comparable to that of the PCR and dengue antibody tests [74]. This report is in concordance with another study that demonstrated that the sensitivity of the NS1 test reached up to 97.4%, with a specificity of 93.7% [75]. Further development of the RAT for dengue fever enables the dengue virus to be distinguished from the Zika virus, which may also be present in serum samples. Monoclonal antibody pairs that have been translated into rapid immunochromatographic tests specifically detect the NS1 protein antigen and distinguish the four dengue virus serotypes from the Zika virus without any cross-reaction. The sensitivity and specificity of this RAT for the detection of dengue virus serotypes range from 76% to 100%, while the sensitivity and specificity for the detection of Zika virus reach up to 81% and 86%, respectively [76]. Thus, the dengue NS1 RAT is a promising detection tool to complement the currently available antibody test to increase the diagnostic efficiency for dengue infection.

### 5.3. Clinical Application of RATs for Malaria

*Plasmodium* spp. are the infectious agents that cause malaria. The majority of malaria cases are found in countries where the cost-effectiveness and ease of diagnostic tests have become major considerations. The RAT for malaria is an alternative detection method that aids clinicians in promptly and effectively diagnosing and treating individuals suspected to have a malaria infection. Malaria antigens currently targeted by this RAT are those that are abundant in all asexual and sexual stages of the parasites, including HRP-2, pLDH, and *Plasmodium* aldolase [77]. Performance evaluation of the ParaSight F immunochromatographic test, which is a RAT that detects the HRP-2 of *P. falciparum* in blood samples, demonstrated a sensitivity ranging from 77% to 98% when >100 parasites/µL were present (0.002% parasitemia). Meanwhile, the specificity ranged from 83% to 98% [77,78,79,80,81,82].

### 5.4. Clinical Application of RATs for Sexually Transmitted Infection

Sexually transmitted infections (STIs) are among the most common infectious diseases with high transmissibility and are significantly associated with morbidity and mortality worldwide [83,84]. According to the WHO, more than one million STIs are acquired every day worldwide, and the majority of those cases are asymptomatic [85]. Rapid detection and point-of-care testing potentially control and prevent the transmission of STIs as well as prevent the sequela of untreated infection. Several point-of-care tests including antibody detection and antigen detection kits have been developed to address this issue, such as rapid antigen tests for the detection of Chlamydia, Neisseria gonorrhoeae, Trichomoniasis vaginalis, and human immunodeficiency virus (HIV) [86,87].

As one of the available rapid antigen tests for Chlamydia infection, the *QuickStripe™* Chlamydia Ag (Savyon Diagnostics Ltd., Ashdod, Israel), a rapid chromatographic immunoassay for the qualitative detection of *Chlamydia trachomatis* in the female cervical swab, male urethral swab, and male urine specimens [88] was reported to be effective, particularly in a high-risk population. With a turnaround time of 10 min, the QuickStripe™ Chlamydia Ag demonstrated a sensitivity of 73.6% and specificity of 81.82%, with positive and negative predictive values of 77.78% and 78.05%, respectively [89]. A systematic review on the clinical performance of RATs for Chlamydia infection, however, reported a pooled sensitivity of 53%, 37%, and 63% for cervical swabs, vaginal swabs, and male urine, respectively, although the specificity was considered high (99%, 97%, and 98%, respectively) [90]. In contrast, according to this review, the aQcare Chlamydia TRF kit (Medisensor, Gyeongsangbuk-do, Korea)*,* a fluorescent nanoparticle-based lateral flow assay, was considered outstanding for the detection. The aQcare Chlamydia TRF kit demonstrated high sensitivity in detecting C. trachomatis with a sensitivity of 93%, a specificity of 96.3%, a positive predictive value of 89.9%, and a negative predictive value of 97.5% [91].

The RATs for the detection of Neisseria gonorrhoeae, including Biostar Opical Immunoassay-Gonorrhea (Biostar Inc., New Taipei, Taiwan), were reported to be promising for point-of-care diagnosis of *N. gonorrhoeae* infection. A laboratory-based evaluation of the Biostar Optical Immunoassay-Gonorrhea reported a high positive detection of up to 99.4% when used to assess the *N. gonorrhoeae* isolates [92]. A pilot study of clinical use of the Biostar Optical Immunoassay-Gonorrhea for the detection of *N. gonorrhoeae* in symptomatic individuals demonstrated a sensitivity of 100% and specificity of 98% compared with NAAT, with 30 min of turnaround time [93].

Subsequently, a rapid antigen testing for the detection of Trichomonas vaginalis demonstrated sensitivity that is comparable to the transcription-mediated amplification (TMA) testing (92.5% and 97.5% for RAT and TMA, respectively) when being used for detection in women who had vaginosis symptoms [94]. A clinical performance study on the Xenotope diagnostic kit (Xenotope Inc, San Antonio, TX, USA), a RAT for detecting *T. vaginalis* specific antigen, demonstrated a short turnaround time (10 min), with a sensitivity and specificity of 90% and 92.5%, respectively, compared to the culture detection method [95].

With a robust development in technology, presently, the RATs for STIs are not only used for detecting a single analyte. A study reported a single RAT that enables the detection of the HIV p24 antigen and the HIV-1 and HIV-2 antibodies. The overall sensitivity and specificity of the combo RAT for HIV diagnosis were 90.5% and 99.8%, respectively, with a turnaround time stated within 45 min [96]. In concordance with this report, a performance evaluation of the Determine™ HIV-1/2 Combo test, a fourth-generation HIV screening assay, demonstrated 100% antibody sensitivity and specificity, and 86.6% antigen sensitivity [97].

## 6. Obstacles to the Application of RATs as a Detection Tool

Despite all the advantages of RATs as detection tools, issues regarding inconsistent results of RATs when being used under certain conditions remain a major shortcoming for the clinical application of RATs as detection tools. User-related errors during sample collection or the interpretation of test results, the condition of the samples, the timing of testing, and the conditions during the distribution and storage of RATs are some of the factors that are prone to induce biases and may hinder the accuracy of RATs (Table 3).

Apart from the previously mentioned factors, an independent evaluation of the RAT for malaria reported that the poor performance of the RAT was related to the possibility of mutation in the gene of the target antigen, cross-reactions of antigens, and susceptibility to heat and humidity [112].

The antibody embedded in the nitrocellulose membrane of RATs is specific to a particular antigen. Any mutation in the gene encoding this antigen may hinder the recognition and binding of an antigen and antibody, hence hampering the accuracy of the RAT. A similar case has been studied and reported in a RAT for SARS-CoV-2. Mutation in the N protein (SARS-CoV-2 antigen that is used as the target analyte in the RAT), particularly at amino acid 135 (T135I), led to discordantly negative and positive results for the Panbio COVID-19 rapid antigen test and RT-PCR test [113].

In concordance with this study, another mutation that is present in the N protein of a SARS-CoV-2 isolate (D399N) was reported to reduce the sensitivity of the Quidel Sofia SARS Antigen FIA Test up to 1000-fold. Interestingly, this mutation did not affect the sensitivity of two other RATs (BinaxNow and Quidel Quickvue antigen tests) [114]. Considering the importance of mutations in the antigen to the overall performance of the RAT, a proper investigation of the relatively conserved antigen as a target analyte and the pairing antibody is necessary during the development of the RAT.

Furthermore, the confidence of the RAT data depends on the accuracy and precision of the antibody to distinguish between the analyte and other structurally similar components. The cross-reactivity that leads to the inaccuracy of the RAT occurs when an antibody for one specific antigen binds to a different antigen that may be present in the sample mixture [115]. Hence, it is very important to ensure that the most specific, high-affinity antibody–antigen complex is chosen for the RAT system.

Finally, compared to molecular testing, such as PCR, the performance of RATs is often less sensitive for samples with a low density of infectious agents. Hence, for high-risk individuals or high-risk populations, when the RAT result comes out negative, it is suggested to repeat the test within 48 h or carry out a confirmation through molecular testing such as PCR before making treatment decisions to prevent the possible spread of the infectious agents due to a false negative.

## 7. Limitation of RATs

Reflecting on the COVID-19 pandemic, RATs have become an important detection tool for clinical diagnosis. Before the COVID-19 era, RATs had also been widely implemented for multiple infectious diseases, as discussed in the section above. RATs offer an efficient alternative to the “gold standard” test to detect individuals with infections, thus proving beneficial in limiting the transmission and accelerating the management of the infected individuals [116]. However, it is noteworthy to mention that RATs have several limitations that should be weighed before being implemented.

One of the limitations of RATs is related to the person who conducted the test. Although procedures to conduct RATs are relatively simple, the feasibility of the inexperienced person incorrectly collecting or handling the samples is likely to be higher than experienced and well-trained personnel. A specimen collection and a test conducted incorrectly may lead to a false or ambiguous test result, which subsequently leads to the poor performance of RATs [117]. For RATs that require relatively easy-to-collect samples such as saliva/sputum, nasal swab, or urine, the outcome of self-testing by a nonprofessional person may not be significantly different compared to the one conducted by a professional [40]. However, nasopharyngeal swab samples or blood samples may require a professional to collect, handle, and conduct the test for an accurate result. Thus, for the antigen test that is intended for self-assessment, instructional resources/information for the correct administration of the RAT should be provided to maximize test performance.

The frequency of false negatives/false positives may also become another limitation of RATs. The occurrence of false negatives/false positives may be due to multiple reasons, e.g., human error, failure to follow the appropriate procedures, a lower load of infectious agents, insufficient clinical specimens, antigen degradation, poorly specific RAT, detection of inactive or residual of infectious agents, or cross-reaction with other substances [118]. Hence, in a situation where clinical suspicion is high, regardless of the initial negative or positive results, the test should be repeated or confirmed by more sensitive assays to establish an accurate diagnosis.

Additionally, RATs are generally generated for the detection of a specific variant of the target infectious agents, and thus may be less sensitive to a different variant of the infectious agents. A study in Ethiopia reported that the *Plasmodium falciparum* parasite lacking histidine-rich protein 2 (*pfhrp2*) and 3 (*pfhrp3*) genes escaped detection by two malaria RATs: CareStart *Pf*/*Pv* (Access Bio, Somerset, NJ, USA) and SD Bioline Malaria Ag P.f. (Abbott Laboratories, North Chicago, IL, USA) [119]. Correspondingly, the study on the clinical performance of COVID-19 RATs also reported impaired detection of the Omicron variant by the first generation of SARS-CoV-2 RATs [120]. Therefore, increasing awareness of the possible reduced detection rate of RATs for different variants is important. Furthermore, providing adequate information about the dominant variant of concerns (VOCs) and the RATs that are relatively sensitive for those VOCs is crucial to address the issue regarding decreased sensitivity of RATs to the VOCs.

## 8. Conclusions

The application of RATs offers advantages in decreasing the laboratory’s burden and shortage of reagents, experimental instruments, or well-trained personnel during mass screening testing. Nevertheless, RATs are tests for infectiousness and thus are not intended to assess the prevalence of infections in a population but to mitigate and prevent further person-to-person transmission. Additionally, RATs are not a replacement for molecular testing, such as PCR or serological testing, but a complement that can be used to aid the screening of infected individuals.

To date, RATs have proven to be a valuable diagnostic tool for detecting not only SARS-CoV-2 but also various infectious agents, making RATs useful to aid in the clinical diagnosis of diseases having high transmissibility. Additionally, for countries with limited resources and access to diagnostic services, RATs may be a good option for quickly identifying the most contagious people; hence, they may prevent the further spread of diseases and aid healthcare workers in making an appropriate decision for treatment.

To address the sensitivity and specificity issues, careful consideration is required for the implementation of RATs in low-prevalence settings. For instance, rapid antigen testing during the early incubation period, when the accumulation of the infectious agents is still low, has to be conducted frequently if the result comes out negative. Alternatively, RAT results have to be confirmed by PCR or other gold-standard methods before making treatment decisions to prevent possible incorrect diagnoses of infectious diseases due to false negatives.

Finally, the performance of RATs can also vary depending on how the individual takes swabs and handles the samples. Ordinary people may find some difficulties in collecting samples and conducting the assessment compared to well-trained healthcare workers. Therefore, proper communication in transferring information regarding the guidelines of how and when to use RATs for non-expert users is also required for home testing or self-testing.

## Figures and Tables

**Figure 1 bioengineering-10-00322-f001:**
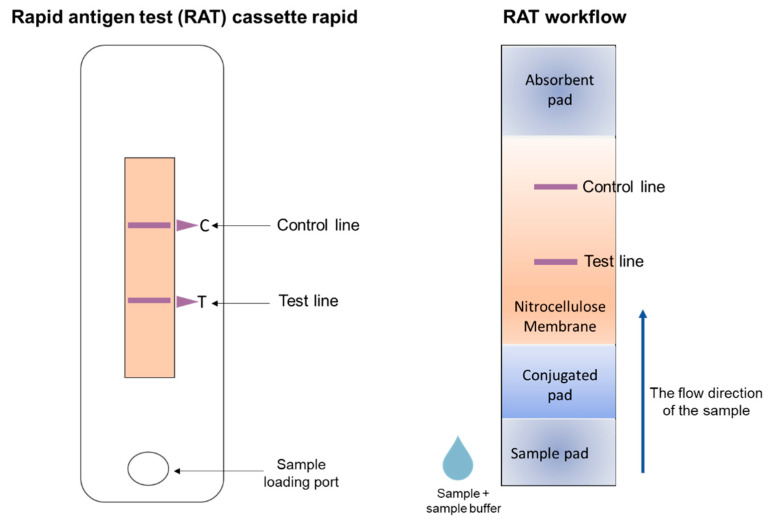
A schematic depicting a rapid antigen test (RAT) that adopts lateral flow immunochromatographic assay technology. The sample–buffer mixture migrates from the sample pad to the conjugated pad where the gold-labeled SARS-CoV-2 protein’s specific antibody is located to form an antibody–antigen complex. Subsequently, the complex migrates to the nitrocellulose detection membrane that contains immobilized antibodies. The lines that appear on both the C and T lines indicate a positive result. The C line must be present for a test to be considered valid.

**Figure 2 bioengineering-10-00322-f002:**
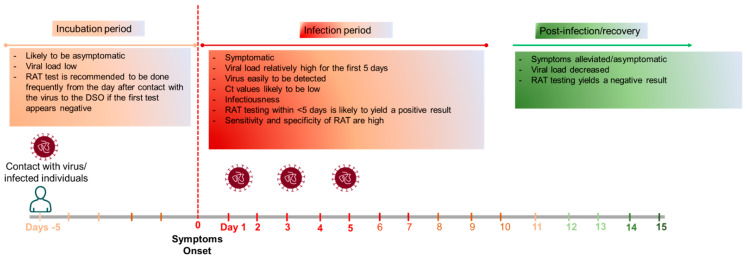
Disease progression of COVID-19 and its influence on RAT performance. The progression of COVID-19 disease is divided into three main periods. In the incubation period (first day of contact with the virus to the day of symptom onset), the viral accumulation is relatively low, and rapid antigen testing within this period is likely to yield a negative result; thus, the test is recommended to be taken frequently. A RAT conducted within 5 days of symptom onset is likely to return a positive result. The performance (sensitivity and specificity) of the RAT is at its highest during this infection period. Eleven days after symptom onset (DSO), the symptoms are likely to become milder to asymptomatic, the viral load decreases, and the RAT during this period will likely return a negative result. If the test remains positive, another test is recommended to be conducted frequently for a certain period (e.g., every two days) until the result is negative.

**Table 1 bioengineering-10-00322-t001:** Commercially available SARS-CoV-2 rapid antigen tests (RATs) in Korea.

No.	RAT Kit Name	Sample Type	Ref.
1	STANDARD^TM^ Q COVID-19 Ag Home Test	Nasal sample	[19,20]
2	GenBody COVID-19 Ag	Nasal sample	[19,21]
3	STANDARD^TM^ F COVID-19 Ag FIA	Nasopharynx sample	[19,22]
4	BIOCREDIT COVID-19 Ag	Nasal sample	[19,23]
5	Humasis COVID-19 Ag Test	Nasopharyngeal or swab sample	[19,24]
6	STANDARD^TM^ i-Q COVID-19 Ag Home Test	Nasal sample	[25,26]
7	MDx COVID-19 Ag Home Test	Nasopharyngeal sample	[25,27]
8	PCL SELF TEST-COVID-19 Ag	Saliva	[25,28]
9	CareStart^TM^ COVID-19 Antigen Home Test	Nasal sample	[25,29]
10	Panbio^TM^ COVID-19 Ag Self-Test	Nasal sample	[25,30]
11	Boditech Quick^TM^ COVID-19 Ag Saliva	Saliva	[25,31]
12	InstaView COVID-19 Antigen Home Test	Nasopharyngeal sample	[25,32]

**Table 2 bioengineering-10-00322-t002:** Characteristics of RT-PCR and RATs for COVID-19 testing.

No.	Characteristics	RT-PCR	RAT
1	Type of analyte that is being assessed	Viral RNA	Viral protein (antigen)
2	Specimens	Nasopharyngeal swab, nasal swab, or saliva	Nasopharyngeal swab, nasal swab, or saliva
3	Sensitivity and specificity		
	- During the incubation period	Moderate	Low
	- During the infection period	High	High
	- During the post-infection period	Moderate	Moderate to low
4	The complexity of the test	Requires complex procedures and specific instruments	Relatively easy to perform
5	Enables point-of-care testing	Some enable the point-of-care testing	Yes
6	Requires a well-trained person to conduct the test	Yes	No
7	Turnaround time	Varies, ranging from a few hours to 1 day	About 15–30 min
8	Suitable for screening	Varies; some may be suitable for screening purposes	Yes
9	Cost	Relatively high cost	Low cost
10	When should the test be used?	- When individuals start developing typical symptoms - When individuals are asymptomatic but have had close contact with infected individuals or are in high-risk regions - 3–5 days after close contact with someone with COVID-19	- When individuals start developing typical symptoms but do not have any access to PCR tests - In the early days after symptom onset (<5 days after DSO) - After the COVID-19 symptoms get better/the individual has no symptoms

Abbreviations: PCR—polymerase chain reaction; RAT—rapid antigen test; RNA—ribonucleic acid; DSO—days after symptom onset.

**Table 3 bioengineering-10-00322-t003:** The factors that affect the performance of RATs.

No.	Factors	Effect on the RAT	Ref.
1	Specimens	- Proper specimens (saliva, nasal/nasopharyngeal swabs, blood, or urine) have to be specifically applied for each RAT. - Specimens from different locations may also contain different amounts of infectious agents (lower or higher densities of infectious agents).	[98,99,100,101]
2	Sample condition	The RATs may perform poorly when being used to assess stored specimens rather than fresh specimens.	[5,46]
3	Sample collection	User-related errors that may occur during the RAT are related to the sample collection. An inaccurate sampling procedure due to a lack of experience may result in poor performance of the RAT.	[5,102]
4	Timing of test	RATs performed within the early days after symptom onset have high positive detection rates.	[5,103,104]
5	Viral load	A high viral load will yield a high detection rate; meanwhile, a lower viral load (insufficient for detection) will yield a low detection rate of the RAT.	[105,106,107]
6	Ct value	Higher Ct values tend to return poor detection rates of RATs.	[58,108]
7	Symptomatic/asymptomatic	RATs have a relatively poor performance when used for asymptomatic individuals.	[109,110,111]
8	RAT handling and storage	Locations with high humidity and temperature or windy conditions rapidly degrade the nitrocellulose capillary flow and thus may affect the performance of the RAT.	[77]

## Data Availability

Not applicable.

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
