# Peer review of "Rapid Antigen Tests during the COVID-19 Era in Korea and Their Implementation as a Detection Tool for Other Infectious Diseases"

_bioengineering, 2023, doi:10.3390/bioengineering10030322_

Round 1
Reviewer 1 Report
This paper analyses different rapid antigen tests (RATs) as diagnostic tools for different infectious agents (SARS-CoV-2, Streptococcus, dengue fever, malaria). Therefore, the title is not in agreement with the content of the paper, or it should be explained: why the detection of malaria has specificities during the COVID Era ? it is not evident for the reviewer.
As expected, most important part of the paper analyses SARS-CoV-2 RATs, which are new assays; However, they limited their study to korean reagents (12 are reported in table 1). This should be mentionned in the title or in the summary. Additionnaly, the performances of only 3 of them are reported and discussed. This should be justified.
- Terms used for the selection of the paper in PubMed and Google Scholar databases should be indicated in section "2. Database search strategy"; The reviewer is surprised by the limited number of papers obtained usinng both bases.
- In table 2, the authors indicate that there is no eanables point of care testing using RT-PCR. It is really surprising, there is a lot rapid tests based on RT-PCR such as products commercialized by Cepheid, Abbott, and so on. This hould be modified. In addition, it is indicated that the turnaround time is mostly 1-2 days. It is not exact, most of the time (at least in my institution), RT-PCR results are available in 12 hours, and my institution (in Europe) is not an exception...
- it should be mentionned that most of the studies performed for the determination of sensitivity and specificity of RATs are performed by traines technicians, and that the performances should be affected when RAT are used by the patients themselves, independantly of the collection of the sample, which of course is a strong limitation.
Author Response
The authors would like to thank the Reviewers for their specific and helpful comments on the manuscript. The authors have carefully considered the comments and have revised the manuscript to address the Reviewers’ concerns. In this revised version, the authors have edited the title and added information regarding the keywords that are being used for reference searching. We also added information regarding the RATs for other infectious diseases, as well as added the limitation of RATs and the possible problem-solving to address each of the obstacles and limitations of RATs as a detection tool to aid diagnosis procedure.
Detail responses to the Reviewers are reported as follows:
Reviewers 1:
- This paper analyses different rapid antigen tests (RATs) as diagnostic tools for different infectious agents (SARS-CoV-2, Streptococcus, dengue fever, malaria). Therefore, the title is not in agreement with the content of the paper, or it should be explained: why the detection of malaria has specificities during the COVID Era ? it is not evident for the reviewer.
Response:
In this article, we intended to discuss the implementation of RAT during the COVID-19 era, particularly in Korea, and the implementation of RATs as a detection tool for other infectious diseases non-COVID-19 including other respiratory diseases non-COVID-19, malaria, dengue fever, and sexually transmitted infection. To address the author’s comment, we amended the title and presented the clarification of the reason for the discussion of the RATs application for the detection of other infectious diseases in the abstract and in the appropriate section.
Title: Rapid Antigen Tests during the COVID-19 Era in Korea and their Implementation as a Detection tool for other Infectious Diseases. (Page 1, Title section).
Abstract:
“Here, the authors review the application of RATs as detection tools for COVID-19, particularly in Korea, as well as for several other infectious diseases. To address these issues, we present general knowledge on the design of RATs that adopt the lateral flow immunoassay for the detection of the analyte (antigen). The authors then discuss the clinical utilization of the authorized RATs amidst the battle against the COVID-19 pandemic in Korea and their role in comparison with the other detection methods. We also discussed the implementation of RATs for other non-COVID-19 infectious diseases, the challenges that may arise during the application, the limitations of RAT as clinical detection tools as well as the possible problem-solving for those challenges to maximize the performance of RATs and avoiding any misinterpretation of the test result.
Page: 1, Lines: 19-27.
Clarification within the manuscript:
Section 5: “RATs as a detection tool to aid the clinical diagnosis of other infectious diseases”
“The application of RATs as a detection tool to aid the clinical diagnosis is not only limited to COVID-19 disease. The application of RATs for the detection of pathogens including influenza viruses, respiratory syncytial virus (RSV), Streptococcus pneumoniae, Streptococcus pyogenes, dengue virus, malaria parasite, and infectious agents that cause multiple sexually transmitted infections has been studied, and reported by multiple groups. The ultimate goal of pathogen detection tools is to aid physicians in making a timely diagnosis and managing the patient properly. Thus, RATs with good sensitivity and specificity are good potential candidates for these purposes when applied at the proper time and in the appropriate procedures. In the sections below, we further discuss the clinical application of RATs for the detection of pathogens that cause diseases other than COVID-19.”
Page: 9, Line: 356
Page: 10, Lines: 366-376.
- As expected, most important part of the paper analyses SARS-CoV-2 RATs, which are new assays; However, they limited their study to korean reagents (12 are reported in table 1). This should be mentionned in the title or in the summary. Additionally, the performances of only 3 of them are reported and discussed. This should be justified.
Response:
As suggested by the reviewer we have amended the title by pointing out that the COVID-19 RATs reagents that are being discussed are the ones that are available in Korea.
Additionally, in the previous version of our manuscript, we only included 4 of 12 available RATs because those 4 RATs are the most well-studied in and out of Korea. However, we considered the author’s comments and decided to add the discussion of the other RATs in separate sections. Nevertheless, given that among 12 available RATs, only 7 of them are properly studied by non-manufacturers independent groups either from Korea or from other countries, we only presented the discussion of 7 RATs. We added the explanation to clarify the reason for discussing only these 7 RATs.
Title: Rapid Antigen Tests during the COVID-19 Era in Korea and their Implementation as a Detection tool for other Infectious Diseases. (Page 1, Title section).
Clarification within the manuscript:
“Although not all commercially available COVID-19 RATs have been sufficiently evaluated under real-world conditions, in the following section, we discuss seven of the twelve authorized RATs in Korea that have been clinically evaluated by several independent study groups in and outside of Korea. As for the other five authorized RATs in Korea, their clinical evaluation report is very limited. The sensitivity and specificity information of these five RATs are limited only to those provided by the manufacturers. “ (Page: 5, Lines: 164 – 169).
Additional discussion for other well-studied RATs:
Pages: 5-7, Lines: 199 - 224 and 246 - 257.
- Terms used for the selection of the paper in PubMed and Google Scholar databases should be indicated in section "2. Database search strategy"; The reviewer is surprised by the limited number of papers obtained using both bases.
Response:
As suggested by the reviewer, we have added the keywords used for the selection of the paper in the PubMed and Google Scholar databases. We also added references to support the discussion in this article.
“For this purpose, the authors searched and identified the relevant publications in the PubMed and Google Scholar databases, as well as online articles that covered the above-mentioned topics as of 2 February 2023 with the keywords “COVID-19 diagnosis”, “rapid antigen test”, “authorized RATs in Korea”, “detection of infectious diseases”, “antigen test for infectious diseases”, “sensitivity of RATs”, “specificity of RATs”, “clinical performance of RAT”, “antigen test for respiratory diseases”, “RAT for sexually transmitted infections” and “limitation of RATs”.”
Page: 2, Lines: 84 – 90.
References section (Pages: 16-21)
- In table 2, the authors indicate that there is no enables point of care testing using RT-PCR. It is really surprising, there are a lot rapid tests based on RT-PCR such as products commercialized by Cepheid, Abbott, and so on. This hould be modified. In addition, it is indicated that the turnaround time is mostly 1-2 days. It is not exact, most of the time (at least in my institution), RT-PCR results are available in 12 hours, and my institution (in Europe) is not an exception...
Response:
We have amended the information for the RT-PCR in table 2 as suggested by the reviewer.
|
No |
Characteristics |
RT-PCR |
RAT |
|
5 |
Enables point-of-care testing |
Some enable the point-of-care testing |
Yes |
|
6 |
Requires a well-trained person to conduct the test |
Yes |
No |
|
7 |
Turnaround time |
Vary, ranging from a few hours to 1 day |
About 15-30 minutes |
Table 2, Page: 9, Lines: 353 – 355.
- it should be mentionned that most of the studies performed for the determination of sensitivity and specificity of RATs are performed by traines technicians, and that the performances should be affected when RAT are used by the patients themselves, independantly of the collection of the sample, which of course is a strong limitation.
Response:
As suggested by the reviewer, we have added a discussion regarding personnel who conducted the test (trained technicians vs self-tester) as one of the RAT limitations.
“One of the limitations of RAT is related to the person who conducted the test. Although procedures to conduct RAT test are relatively simple, the feasibility of the inexperienced person incorrectly collecting or handling the samples are likely to be higher than experienced and well-trained personnel. A specimen collection and a test conducted incorrectly may lead to a false or ambiguous test result, which subsequently leads to the poor performance of RATs. For RATs that require relatively easy-to-collect samples such as saliva/sputum, nasal swab, or urine, the outcome of self-testing by a nonprofessional person may not be significantly different compared to the one conducted by a professional. However, nasopharyngeal swab samples or blood samples may require a professional to collect, handle and conduct the test for an accurate result. Thus, for the antigen test that is intended for self-assessment, instructional resources/information for the correct administration of the RAT test should be provided to maximize test performance.”
Section 7 (Limitation of RATs)
Page: 14, Lines: 587 - 598
Reviewer 2 Report
1. What is the novelty of this article while similar articles are already published?
https://www.ncbi.nlm.nih.gov/pmc/articles/PMC8234251/
2. What is the impact or contribution of this review article to the respective scientific field? Please discuss and update the abstract as well.
3. Authors talk about respiratory infectious diseases while there is no information about the rapid antigen tests for tuberculosis which are available and discussed in the published articles. Please include all the relevant information.
4. Rapid antigen tests are available for many other infections such as vaginosis, however, the article missing information. Please include and update the article with all available rapid antigen tests for infectious diseases.
5. What are the limitations of RATs? Please include a separate section for the limitations.
Author Response
The authors would like to thank the Reviewers for their specific and helpful comments on the manuscript. The authors have carefully considered the comments and have revised the manuscript to address the Reviewers’ concerns. In this revised version, the authors have edited the title and added information regarding the keywords that are being used for reference searching. We also added information regarding the RATs for other infectious diseases, as well as added the limitation of RATs and the possible problem-solving to address each of the obstacles and limitations of RATs as a detection tool to aid diagnosis procedure.
Detail responses to the Reviewers are reported as follows:
Reviewer 2:
- What is the novelty of this article while similar articles are already published?
https://www.ncbi.nlm.nih.gov/pmc/articles/PMC8234251/
Response:
Although the presentation in our manuscript may be similar to the one pointed out by the reviewer, however, in our manuscript, we not only presented the implementation of RATs either for COVID-19 or other infectious diseases but also presented the obstacles for the implementation of RATs, as well as the major limitation of RATs and the possible problem solving to address each of those obstacles, thus may help to maximize the performance of RATs when being implemented.
- What is the impact or contribution of this review article to the respective scientific field? Please discuss and update the abstract as well.
Response:
With this review, we aimed to give new insight for the readership to easily understand the benefits and limitations of RATs. We also discuss and offer possible problem-solving for obstacles or limitations of RATs. For instance, we discussed how the RATs are supposed to be used and the most effective period to conduct the RAT test. We thought these are important, given most of the discrepancies in RATs results were due to the inaccurate use of RATs and the period for the test.
Possible problem-solving to maximize the performance of RAT presented in the manuscript:
“Considering the importance of mutations in the antigen to the overall performance of the RAT, a proper investigation of the relatively conserved antigen as a target analyte and the pairing antibody is necessary during the development of the RAT.” (Page 14, 563-566).
“Furthermore, the confidence of the RAT data depends on the accuracy and precision of the antibody to distinguish between the analyte and other structurally similar components. The cross-reactivity that leads to the inaccuracy of the RAT occurs when an antibody for one specific antigen binds to a different antigen that may be present in the sample mixture. Hence, it is very important to ensure that the most specific, high-affinity antibody–antigen complex is chosen for the RAT system.
Finally, compared to molecular testing, such as PCR, RATs’ performance is often less sensitive for samples with a low density of infectious agents. Hence, for high-risk individuals or high-risk populations, when the RAT result comes out negative, it is suggested to repeat the test within 48 hours or carry out a confirmation through molecular testing such as PCR before making treatment decisions to prevent the possible spread of the virus due to a false negative.” (Page 14, Line 567 -578).
“However, nasopharyngeal swab samples or blood may require a professional to collect, handle and conduct the test for an accurate result. Thus, for the antigen test that is intended for self-assessment, instructional resources/information for the correct administration of the RAT test should be provided to maximize test performance. “(Page 14, Lines: 595-599)
“The frequency of false negative/false positive form RATs tests may also become another limitation of RATs. The occurrence of false negative/false positive may be due to multiple reasons, i.e., human error, failure to follow the appropriate procedures, a lower load of infectious agents, insufficient clinical specimens, antigen degradation, poorly specific RAT, detection of inactive or residual of infectious agents, or cross-reaction with other substances. Hence, in a situation where clinical suspicion is high, regardless of the initial negative or positive results, the test should be repeated or confirmed by more sensitive assays to establish an accurate diagnosis.” (Page 15, Lines: 600-607).
“Therefore, it is important to increase awareness of the possible reduced detection rate of RATs for different variants. Furthermore, providing adequate information about the dominant variant of concerns (VOCs) and the RATs that are relatively sensitive for those VOCs is crucial to address the issue regarding decreased sensitivity of RATs to the VOCs.” (Page 15, Lines: 615 – 619).
In the Conclusion section:
“To address the sensitivity and specificity issues, careful consideration is required for the implementation of RATs in low-prevalence settings. For instance, rapid antigen testing during the early incubation period, when the accumulation of the infectious agents is still low, has to be conducted frequently if the result comes out negative. Alternatively, RAT results have to be confirmed by PCR or other gold standard methods before making treatment decisions to prevent possible incorrect diagnoses of infectious diseases due to false negatives.
Finally, the performance of RATs can also vary depending on how the individual takes swabs and handles the samples. Ordinary people may find some difficulties in collecting samples and conducting the assessment compared to well-trained healthcare workers. Therefore, proper communication in transferring information regarding the guidelines of how and when to use RATs for non-expert users is also required for home testing or self-testing. “ (Page: 15, Lines: 633 – 639)
- Authors talk about respiratory infectious diseases while there is no information about the rapid antigen tests for tuberculosis which are available and discussed in the published articles. Please include all the relevant information.
Response:
As suggested by the reviewer, we have added the discussion of the antigen test for tuberculosis.
“Additionally, antigen detection test was also widely used for the detection of Mycobacterium tuberculosis, bacteria that causes tuberculosis diseases. Tuberculosis is a highly transmitted disease that is also considered one of the major problems in the world. Evaluation of the commercially available RAT for tuberculosis in India demonstrated that the performance of the RAT depended on the bacterial load in the sample. A sample with a high bacterial load yielded a positivity of up to 100%, but with a lower bacterial load, the positivity decreased by up to 12.06%. Another study on the antigen test for tuberculosis, however, reported the potential of two Mycobacterium tuberculosis-specific antigens, named ESAT-6 and CFP10, as analytes of RAT for tuberculosis. The combination of the ESAT-6 and CFP10 was found to be highly sensitive and specific for diagnosis, with a sensitivity of up to 73% and specificity of up to 93%. Correspondingly, the clinical performance evaluation of the tuberculosis RAT developed by the Pasteur Institute of Iran, named PrTBK, demonstrated 100% sensitivity, 89% specificity, and 92% accuracy in comparison with the culture method. Furthermore, a study in 2019 reported the detection of tuberculosis lipoarabinomannan (LAM) antigen in the urine sample of children with presumed TB. According to this study, the urinary LAM antigen testing enables the point-of-care diagnosis with relatively high sensitivity up to 73.2% in confirmed intrathoracic tuberculosis (ITTB) cases, and 76% in confirmed lymph node tuberculosis (LNTB) cases. The specificity of the LAM-RAT assay in children with ITTB and those with LNTB was 92% and 93%, respectively. ”
(Page 11, Lines: 426-445)
- Rapid antigen tests are available for many other infections such as vaginosis, however, the article missing information. Please include and update the article with all available rapid antigen tests for infectious diseases.
Response:
As suggested by the reviewer, we have added the discussion of the other available rapid antigen tests for other infectious diseases.
“Furthermore, RATs were also reported to be applied for the detection of two bacteria that caused a type of pneumonia, Streptococcus pneumoniae, and Legionella pneumophila. S. pneumoniae is the most commonly identified infectious agent in community-acquired pneumonia (CAP), whereas L. pneumophila, a causative agent of uncommon but potentially serious pneumonia (legionnaires’ disease), is considered less frequently identified in CAP. Clinical evaluation of four S. pneumoniae urinary antigen tests, the BinaxNOW S. pneumoniae Antigen Card (Abbott, Chicago, IL, USA), ImmuView S. pneumoniae and Legionella (SSI Diagnostica, Hillerød, Denmark), STANDARD F S. pneumoniae Ag FIA (SD Biosensor, Gyeonggi, South Korea), and Sofia S. pneumoniae FIA (Quidel Corporation, San Diego, CA, USA) demonstrated a sensitivity ranged from 76.9% to 86.5%, and specificity that ranging from 84.2% to 89.7%. No significant difference was found among the four test kits. Correspondingly, a comparison of a novel antigen test kit, IMMONOCATCHTM Streptococcus penumoniae (EIKEN CHEMICAL CO., Ltd, Tokyo, Japan) with two other test kits, Uni-GoldTM S. pneumoniae (Trinity Biotech, Bray, Ireland), and BinaxNOW S. pneumoniae (Abbott, Illinois, U.S.A.) demonstrated comparable results. The sensitivity and specificity of the IMMUNOCATCH were 73.2% and 98.9%, respectively, compared to the Uni-Gold. Whereas, when compared to BinaxNOW were 97.6% and 98.8%, respectively. Together, these studies suggest that the RAT is a useful tool for the detection of S. pneumoniae.
Subsequently, the RAT was also widely used for the detection of L. pneumophila. The clinical evaluation of the Legionella K-SeT (CORIS BioConcept, Gembloux, Belgium), an immunochromatographic assay for the detection of L. pneumophila from a urine sample, demonstrated sensitivity, specificity, positive predictive value, negative predictive value, and accuracy of 75.5%, 100%, 100%, 98.7%, and 98.9% respectively. Furthermore, a head-to-head comparison between the ImmuView L. pneumophila and L. longbeachae urinary antigen test (SSI Diagnostica A/S, Hillerød, Denmark) and the BinaxNOW Legionella urinary antigen card (Abbott, Illinois, U.S.A.) also demonstrated high sensitivity and specificity of 96.0% and 100%, respectively, for both RATs when being used to detect the L. pneumophila from the urine samples.”
Pages: 10-11, Lines: 397 – 425
“5.4. Clinical application of RAT for sexually transmitted infection
Sexually transmitted infections (STIs) are among the most common infectious diseases with high transmissibility and are significantly associated with morbidity and mortality worldwide. According to WHO, more than one million STIs are acquired every day worldwide, and the majority of those cases are asymptomatic. Rapid detection and the point-of-care test potentially control and prevent the transmission of STIs as well as prevent the sequelae of untreated infection. Several point-of-care tests including antibodies detection and antigen detection kits have been developed to address this issue, including rapid antigen tests for the detection of Chlamydia, Neisseria gonorrhoeae, Trichomoniasis vaginalis, and human immunodeficiency virus (HIV).
As one of the available rapid antigen tests for Chlamydia infection, the QuickStripe™ Chlamydia Ag, a rapid chromatographic immunoassay for the qualitative detection of Chlamydia trachomatis in the female cervical swab, male urethral swab, and male urine specimens was reported to be effective as an alternative diagnostic test for Chlamydia infection, particularly in a high-risk population. With a turnaround time of 10 minutes, the QuickStripe™ Chlamydia Ag demonstrated a sensitivity of 73.6% and specificity of 81.82%, with positive and negative predictive values, of 77.78% and 78.05%, respectively. A systematic review on the clinical performance of RAT for Chlamydia infection, however, reported a pooled sensitivity of 53%, 37%, and 63% for cervical swabs, vaginal swabs, and male urine, respectively, although the specificity was considered high (99%, 97%, and 98%, respectively). In contrast, according to this review, the aQcare Chlamydia TRF kit, a fluorescent nanoparticle-based lateral flow assay was considered outstanding for the detection. The aQcare Chlamydia TRF kit demonstrated high sensitivity in detecting the Chlamydia trachomatis with a sensitivity of 93%, specificity of 96.3%, positive predictive value of 89.9%, and negative predictive value of 97.5%.
The RAT tests for detection of Neisseria gonorrhoeae including Biostar Opical Immunoassay-Gonorrhea (Biostar, Inc) were reported to be promising for point-of-care diagnosis of N. gonorrhoeae infection. A laboratory-based evaluation of the Biostar Optical Immunoassay-Gonorrhea reported a high positive detection of up to 99.4% when used to assess the N. gonorrhoeae isolates. A pilot study of clinical use of the Biostar Optical Immunoassay-Gonorrhea for the detection of N. gonorrhoeae in symptomatic individuals demonstrated a sensitivity of 100% and specificity of 98% compared with NAAT, with 30 minutes of turnaround time.
Subsequently, a rapid antigen testing for the detection of Trichomonas vaginalis demonstrated sensitivity that is comparable to the transcription-mediated amplification (TMA) testing (92.5% and 97.5% for RAT and TMA, respectively) when being used for detection in women who had vaginosis symptoms. Clinical performance study on the Xenotope diagnostic kit (Xenotope Inc, San Antonio, USA), a RAT for detecting T. vaginalis specific antigen demonstrated a short turnaround time (10 minutes), with sensitivity and specificity of 90% and 92.5%, respectively, compared to the culture detection method.
With a robust development in technology, presently, the RATs for STIs are not only used for detecting a single analyte. A study reported a single RAT that enables the detection of the HIV p24 antigen and the HIV-1 and HIV-2 antibodies. The overall sensitivity and specificity of the combo RAT for HIV diagnosis were 90.5% and 99.8%, respectively, with a turnaround time stated within 45 minutes. In concordance with this report, a performance evaluation of the Determine™ HIV-1/2 Combo test, a fourth-generation HIV screening assay demonstrated 100% antibodies sensitivity and specificity, and 86.6% antigen sensitivity.
Page 11-13, Lines: 491-539
- What are the limitations of RATs? Please include a separate section for the limitations.
Response: As suggested by the reviewer, we have added an independent section that discussed the limitations of RATs.
“ 7. Limitation of RATs
Reflecting on the COVID-19 pandemic, the RATs have become an important detection tool for clinical diagnosis. Before the COVID-19 era, the RATs had also been widely implemented for multiple infectious diseases, as discussed in the section above. The RATs offer an efficient alternative to the ‘gold standard’ test to detect individuals with infections, thus giving beneficial in limiting the transmission and accelerating the management of the infected individuals [115]. However, it is noteworthy to mention that RATs have several limitations that should be weighed before being implemented.
One of the RAT’s limitations is related to the person who conducted the test. Although procedures to conduct RAT test are relatively easy, the feasibility of the inexperienced person incorrectly collecting or handling the samples are likely to be higher than experienced and well-trained personnel. A specimen that is incorrectly collected and a test conducted incorrectly may lead to a false or ambiguous test result, which subsequently will lead to the poor performance of RATs [116]. For RATs that require relatively easy-to-collect samples such as saliva/sputum, nasal swab, or urine, the outcome of self-testing by a nonprofessional person may not be significantly different compared to the one conducted by a professional [40]. However, nasopharyngeal swab samples or blood samples may require a professional to collect, handle and conduct the test for an accurate result. Thus, for the antigen test that is intended for self-assessment, instructional resources/information for the correct administration of the RAT test should be provided to maximize test performance.
The frequency of false negative/false positive form RATs tests may also become another limitation of RATs. The occurrence of false negative/false positive may be due to multiple reasons, i.e., human error, failure to follow the appropriate procedures, a lower load of infectious agents, insufficient clinical specimens, antigen degradation, poorly specific RAT, detection of inactive or residual of infectious agents, or cross-reaction with other substances [117]. Hence, in a situation where clinical suspicion is high, regardless of the initial negative or positive results, the test should be repeated or confirmed by more sensitive assays to establish an accurate diagnosis.
Additionally, RATs are generally generated for the detection of a specific variant of the target infectious agents, and thus may be less sensitive to a different variant of the infectious agents. A study in Ethiopia reported that the Plasmodium falciparum parasite lacking histidine-rich protein 2 (pfhrp2) and 3 (pfhrp3) genes escape the detection by two malaria RATs: CareStart Pf/Pv (Access Bio, Belgium) and SD Bioline Malaria Ag P.f. (Abbott, USA) [118]. Correspondingly, the study on the clinical performance of COVID-19 RAT also reported impaired detection of the Omicron variant by the first generation of SARS-CoV-2 RATs [119]. Therefore, increasing awareness of the possible reduced detection rate of RATs for different variants is important. Furthermore, providing adequate information about the dominant variant of concerns (VOCs) and the RATs that are relatively sensitive for those VOCs is crucial to address the issue regarding decreased sensitivity of RATs to the VOCs. “
Pages: 14-15
Lines: 580 - 619
Round 2
Reviewer 1 Report
The authors have taken into account my comments. I agree with this new version.
Reviewer 2 Report
Authors successfully responded to the reviewer's comments and updated the manuscript as well.